# Sensitivity of Model-Based Predictions of Post-TKA Kinematic Behavior to Residual Errors in Ultrasound-Based Knee Collateral Ligament Strain Assessment

**DOI:** 10.3390/s23198268

**Published:** 2023-10-06

**Authors:** Félix Dandois, Orçun Taylan, Jacobus H. Müller, Lennart Scheys

**Affiliations:** 1Institute for Orthopaedic Research and Training (IORT), Development and Regeneration Department, KU Leuven, 49 Herestraat, 3000 Leuven, Belgiumorcun.taylan@kuleuven.be (O.T.); 2ReSurg SA, Rue Saint-Jean 22, 1260 Nyon, Switzerland; 3Department of Orthopaedics, University Hospitals Leuven, 49 Herestraat, 3000 Leuven, Belgium

**Keywords:** collateral ligaments, strains, ultrasound, modeling, knee arthroplasty

## Abstract

Ultrasound-based ligament strain estimation shows promise in non-invasively assessing knee joint collateral ligament behavior and improving ligament balancing procedures. However, the impact of ultrasound-based strain estimation residual errors on in-silico arthroplasty predictions remains unexplored. We investigated the sensitivity of post-arthroplasty kinematic predictions to ultrasound-based strain estimation errors compared to clinical inaccuracies in implant positioning.Two cadaveric legs were submitted to active squatting, and specimen-specific rigid computer models were formulated. Mechanical properties of the ligament model were optimized to reproduce experimentally obtained tibiofemoral kinematics and loads with minimal error. Resulting remaining errors were comparable to the current state-of-the-art. Ultrasound-derived strain residual errors were then introduced by perturbing lateral collateral ligament (LCL) and medial collateral ligament (MCL) stiffness. Afterwards, the implant position was perturbed to match with the current clinical inaccuracies reported in the literature. Finally, the impact on simulated post-arthroplasty tibiofemoral kinematics was compared for both perturbation scenarios. Ultrasound-based errors minimally affected kinematic outcomes (mean differences < 0.73° in rotations, 0.1 mm in translations). Greatest differences occurred in external tibial rotations (−0.61° to 0.73° for MCL, −0.28° to 0.27° for LCL). Comparatively, changes in implant position had larger effects, with mean differences up to 1.95° in external tibial rotation and 0.7 mm in mediolateral translation. In conclusion, our study demonstrated that the ultrasound-based assessment of collateral ligament strains has the potential to enhance current computer-based pre-operative knee arthroplasty planning.

## 1. Introduction

Total knee arthroplasty (TKA) is a common orthopedic procedure, representing 1.5 million procedures per year in countries part of the organization for economic cooperation and development alone, and expected to further increase by 2030 [1]. However, TKA still fails in 2–8% of the cases [2,3] and 35% of these failures has been attributed to ligament imbalance, which is associated with excessive stiffness or instability of the knee joint [4]. Nonetheless, current ligament balancing procedures applied during TKA primarily rely on indirect assessments of ligament strains, e.g., bony distances or intra-articular distances, and/or subjective intra-operative opinions of the surgeon [5,6,7]. The two main ligaments targeted by such ligament balancing procedures are the medial collateral ligament (MCL) and lateral collateral ligament (LCL) because they are the primary frontal-plane knee stabilizers post-TKA [8,9]. Therefore, direct assessment of functionally relevant in-situ biomechanical properties, e.g., ligament tension and strain in response to a given varus or valgus stress, could be of great value. Indeed, these properties could provide surgeons with currently unavailable objective data to further improve both the pre-operative surgical planning and intra-operative ligament balancing procedures, thereby reducing the above TKA failures related to ligament imbalance. Additionally, robot-assisted TKA, which already displayed improved stability compared to conventional TKA [10,11], are currently primarily based on soft-tissue properties extracted from the scarce literature available. As a result, they could also benefit from subject-specific biomechanical properties to improve the accuracy of their underlying ligament balancing procedure [12,13].

Ultrasound (US) imaging is a common imaging modality to non-invasively assess dynamic biomechanical properties of soft-tissues such as tendons and ligaments in-situ. Indeed, dedicated ultrasound imaging methodologies, i.e., speckle tracking, have already been developed and successfully applied on large energy-storing tendons, e.g., patellar tendon and Achilles tendon [14,15,16]. However, only a limited number of studies applied these techniques on knee collateral ligaments, primarily due to their inherently complex geometry and dynamic behavior during clinically relevant functional assessments of the knee joint [17]. Additionally, currently available studies were either performed ex-situ on isolated ligaments [18], hence preventing extrapolation to the above clinically relevant functional assessments, or were lacking ground-truth data to thoroughly validate the process [17]. Recently, a study aiming to overcome these limitations was performed by our group [19]. Herein, a US speckle tracking approach was developed to specifically measure in-situ collateral ligament strain, and was successfully validated through the acquisition of reference data using digital image correlation (DIC) during varus–valgus loading, indicating the potential of ultrasound-based ligament strain estimation to non-invasively assess collateral ligament behavior in the knee joint and improved ligament balancing in knee arthroplasty, for example through its integration in computer-based pre-operative planning. Nevertheless, partly due to the novelty of this research area, the impact of residual errors associated with this technique on the in-silico estimation of functionally-relevant knee arthroplasty outcomes have never been tested, thereby hindering the potential clinical applications of that study.

Therefore, this project investigated the sensitivity of model-based predictions of post-TKA tibiofemoral (TF) kinematics to residual errors in US-based collateral ligament strain estimations. In addition, the resulting sensitivity in terms of tibiofemoral kinematics was compared to the effect of the current clinical variability in terms of implant positioning on post-operative TF kinematics. We hypothesized that the sensitivity of model-based predictions associated with the existing US-based strain assessment of in-situ collateral ligaments would be less than the sensitivity to current surgical inaccuracies in terms of implant positioning.

## 2. Materials and Methods

### 2.1. Experimental Data Collection

Two fresh-frozen cadaveric specimens (specimen 1: gender: male, age = 83 years, BMI = 31.21 kg/m^2^; specimen 2: gender: male, age = 87 years old, BMI = 30.09 kg/m^2^) without any sign of lower limb disorders or prior surgical intervention were obtained from the institute’s body donation program, following ethical approval by the local ethics committee (NH019 2017-02-03). For each specimen, full leg radiographs and magnetic resonance images were acquired first. Afterwards, specimens were prepared following a standardized and validated procedure commonly applied for knee-joint simulator experiments at our institution [20,21,22,23]. Rigid marker frames with reflective spheres were attached to the tibia and femur using bicortical bone pins. Afterwards, computed-tomography (CT) images with a slice thickness of 0.6 mm were acquired for each specimen in full extension. Based hereon, marker positions were identified in relation to anatomical landmark locations in Mimics (Mimics 20.0, Materialize, Leuven, Belgium) to define a knee joint coordinate system based on the Grood and Suntay convention [24]. Twenty-four hours prior to the experiments, each specimen was thawed at room temperature and resected 32 cm proximally and 28 cm distally to the knee joint. The skin and subcutaneous tissue around the knee joint were carefully removed while preserving the joint capsule, ligaments and tendons. The tibia and femur were embedded into custom metal pots in a physiological orientation using acrylic resin (VersoCit2, Struers, Ballerup, Denmark). Suture loops were passed through the medial and lateral hamstrings. Exposed quadriceps tendons were secured within a custom-made clamp to affix to an electromechanical actuator for dynamic control [25].

Following specimen preparation, each leg underwent a posterior-stabilized TKA (Genesis II, Smith & Nephew, Memphis, TN, USA) based on the company’s pre-operative surgical plan and materialized using MRI-based subject-specific cutting blocks (VISIONAIRE, Smith & Nephew, Memphis, TN, USA). Surgeons had the possibility to intra-operatively modify the implant size and component alignment, and to perform the ligament release to achieve soft tissue balance based on their experience and opinion [6,7].

Following the surgical procedure, each specimen was mounted on a well-established ex-vivo physiological knee-joint simulator [25]. First, passive flexion–extension of the knee was manually performed to determine the laxity of both specimens, defined as the absolute difference between minimal and maximal kinematic values, across the complete range of motion (Table 1). Second, active squatting was performed by imposing a vertical displacement on the proximal femur to achieve a cyclic knee motion, while applying a dynamic physiological quadriceps load using an embedded electromechanical actuator to generate and maintain a vertical ankle force of 90N. Throughout, the medial and lateral hamstrings were loaded with 50 N constant force springs. Specimens were kept moist with phosphate-buffered saline solution to mitigate tissue-drying effects. During the active squatting motion, the motion of the tibia and femur were tracked using a six-camera motion capture system (capture frequency = 100 Hz, MX40 cameras; Vicon, Oxford, UK).

The recorded trajectories of markers on the femur and tibia were then further processed (Nexus 2.9, Vicon, Oxford, UK) to calculate tibiofemoral kinematics for each specimen using a custom code (MATLAB R2018b, MathWorks Inc, Natick, MA, USA). Upon completion of the experiments, post-operative CT images were again acquired for each specimen in full extension.

### 2.2. Computer Modelling

For each specimen, a rigid-body computational model was formulated in ADAMS (ADAMS 2018.1, MSC Software Corporation, Newport Beach, CA, USA), based on an existing virtual knee-joint simulator modeling pipeline [26,27]. The models integrated specimen-specific bone geometries of femurs, tibias and patellas extracted from the available pre-operative CT data, as well as the implant stereolithography (STL) models including femoral component, tibial baseplate and polyethylene insert. Position and orientations of implant components relative to the bone was defined by manually registering available STL models with post-operative CT scans in 3-Matic (3-Matic, Materialize, Leuven, Belgium).

In addition to the bone and implant geometries, five ligaments were defined: MCL, LCL, medial patella-femoral ligament (MPFL), lateral patella-femoral ligament (LPFL) and patellar ligament. LCL, MPFL and LPFL were all modeled as one linear bundle; MCL was modeled using two linear bundles (anterior and posterior); and the patellar ligament as three bundles (medial, lateral and central). The insertions of all ligaments were precisely determined by referencing the bony landmarks of each specimen, and by using anatomical atlases as guides [28,29]. The ligament forces were defined using the well-known Blankevoort formulation [30]:(1)$$f=\{\begin{array}{cc} 0, & \varepsilon <0\\ \frac{1}{4}k\varepsilon^{2}/\varepsilon_{l}\;, & 0\leq \varepsilon \leq 2\varepsilon_{l}\\ k\left(\varepsilon -\varepsilon_{l}\right)\;, & \varepsilon >2\varepsilon_{l}, \end{array}$$
where *f* is the tensile force, *k* the ligament stiffness, $\varepsilon_{l}$ the linear strain limit, set at 0.03 [30], and *ε* the strain calculated using the equation:(2)$$\varepsilon =\frac{L-L_{0}}{L_{0}},$$
where *L* is the ligament length and $L_{0}$ is the zero-load length computed with the following equation [31]:(3)$$L_{0}=\frac{L_{r}}{\varepsilon_{r}+1}$$

Herein, $L_{r}\;$is the ligament reference length and $\varepsilon_{r}$ the reference strain. Ligament reference length was defined as the distance between the insertions at full length following an equilibrium simulation, wherein the femoral component should be seated properly in the tibial insert with the knee in full extension [26]. In addition to the ligaments, medial (semimembranosus and semitendinosus) and lateral (biceps femoris) hamstrings as well as the quadriceps tendon were modeled. The medial and lateral hamstrings were modeled as a linear constant force vector, while the quadriceps tendon was modeled as three bundles.

The subsequently generated rigid-body models were then integrated in a virtual knee-joint simulator aiming to reproduce the experimental conditions. The force experimentally applied through the quadriceps tendon was defined through the following equation:(4)Fq=Passive quadriceps force+PID+preload,
with Fq being the resulting quadriceps force, the “Passive quadriceps force” is modeled as $\alpha \;*\;\exp\left(\beta \;*\;Quad_{length}\right)\;$with α and β fine-tuned by trial and error for each specimen to obtain quadriceps loads that best match experimentally obtained results (Table 2) [32], “PID” represents the force component originating from the joint simulator’s PID regulation used to maintain a constant 90 N vertical load at the ankle joint. The “preload” component accounts for the preloading of the quadriceps tendons applied experimentally at the starting position of the squatting motion (35° of flexion).

In contrast with prior studies, wherein the virtual knee-joint simulator was directly driven by the quadriceps load [29], this virtual knee-joint simulator was driven by hip displacement to more adequately reproduce the experimental set-up, which is also driven by hip displacement. Furthermore, as in the experimental set-up, the quadriceps load was modeled using a PID regulation, in contrast with quadriceps-load-driven models, hence providing additional data for the validation of the models. Likewise, a second PD regulation was applied to minimize the difference between experimental hip displacement and simulated hip displacement at each time point. Finally, the hip, knee and ankle joint constraints integrated in the experimental set-up [25] were also recreated in the virtual knee-joint simulator (Figure 1). The hip joint was constrained and could translate vertically and rotate in the sagittal plane. The ankle joint was modelled with five degrees of freedom, eliminating anterior–posterior displacement. The TF and patella-femoral (PF) joints were constrained by implant and bone geometries, as well as soft-tissues. The TF and PF contacts were defined using following equation [27,33]:(5)$$F_{cont}=k_{c}\delta_{int}^{\tau }+C_{c}{\dot{\delta }}_{int}^{},$$
where $F_{cont}\;$is the contact force$;\;k_{c}$ is the contact stiffness set at 5000 N/mm; *τ* is the contact exponent set at 2.2, as recommended in the software documentation; $C_{c}$ is the contact damping set at 10 N · s/mm; $\delta_{int}\;$is the amount of overlap; and ${\dot{\delta }}_{int}^{.}\;$is the speed of overlapping between interfacing bodies. These values were extracted from the available literature, in which similar models were developed and validated [27].

To initialize the simulation, reference strain and initial stiffness were likewise extracted from the literature [29,31]. Afterwards, stiffness of all ligaments was fine-tuned to iteratively minimize the differences between simulated and experimental results in terms of loads, i.e., quadriceps and ankle load, and TF kinematics, i.e., the three rotations and the three translations, as well as maximizing their Pearson correlation coefficient. The resulting parameters for all ligaments and each specimen are listed in Table 3.

The fidelity of the rigid-body model and virtual definition of the knee-joint simulator was established through visual inspection of the resulting simulations, screening for possible penetration or unrealistic behavior. Furthermore, we quantified its accuracy by calculating differences between experimentally measured and simulated kinematics and kinetics, as well as the associated Pearson correlation coefficients.

### 2.3. Sensitivity Analysis

Once the ligament parameters were fine-tuned, a sensitivity analysis was performed. First, the sensitivity to residual errors in US-based MCL and LCL strain estimations was studied. This task was performed by iteratively modifying the stiffness of MCL and LCL individually until the simulated differences in strains match previously reported differences between US-based and DIC-based strains, i.e., 0.27% and 0.57% strain for MCL and LCL, respectively [19]. The TF kinematics measured with these simulations will be further referred as US TF kinematics. Mean differences across the complete squat cycle between TF kinematics with the fine-tuned ligament parameters (Table 4) and US TF kinematics with perturbed ligament parameters were then computed.

Secondly, the sensitivity of the models to clinical accuracy in terms of implant positioning was analyzed. This was performed by modifying the internal–external rotation of the tibial base plate and polyethylene insert by ±3.2°, i.e., the previously reported difference between surgical plan and actually achieved alignment [34]. During these simulations, the optimal ligament parameters reported in Table 4 were used. The TF kinematics measured with these simulations will be further referred as IMP TF kinematics. Afterwards, mean differences across the complete squat cycle between optimal TF kinematics and IMP TF kinematics were again computed. Similarly, associated differences between US TF kinematics and IMP TF kinematics were quantified.

### 2.4. Statistics

The translations were reported in the tibial reference frame with positive values assigned to medial, anterior and proximal translations and to valgus and external tibial rotations. The differences between the experimental and simulation results were quantified using root-mean-square error (RMSE) and the Pearson correlation coefficient with ρ categorized as ρ ≤ 0.35, 0.35 < ρ ≤ 0.67, 0.67 < ρ ≤ 0.9, 0.9 < ρ to be weak, moderate, strong or excellent correlations [35]. In addition, motion was visually assessed to detect any abnormal behavior, e.g., bone and implant penetration, dislocation, etc.

## 3. Results

### 3.1. Validity of Computer Simulations

Differences between experimentally measured and simulated kinematics and kinetics as well as the associated Pearson correlation coefficients can be found in Table 4. Graphs of the associated kinematics can also be found in Figure 2. The visual inspection during the complete squat cycle did not reveal any penetration or unrealistic behavior.

### 3.2. Sensitivity Analyses

The MCL and LCL stiffness of each specimen to reach a difference in strains equal to the error observed with US-based collateral ligament strain estimation can be found in Table 5. For Specimen 2, LCL had no impact on the kinematics. Indeed, according to previously defined equations, no force is applied if the ligament length is inferior to reference length during the whole range of motion, which is the case for the LCL of specimen 2.

Average (standard deviation) differences between optimal TF kinematics and US TF kinematics as well as the differences between optimal TF kinematics and IMP TF kinematics can be found in Table 6. Comparisons of varus–valgus and internal–external tibial rotation between US simulation and IMP simulation are displayed on Figure 3.

## 4. Discussion

This study investigated the sensitivity of predicted kinematic behavior post-total knee arthroplasty to previously reported errors in US-based collateral ligament strain estimations. Thereto, a virtual simulator with an integrated knee joint model was created to simulate experimental squatting of two specimens and was validated by comparing simulated TF kinematics and muscle loads with experimental in-vitro results. These validated reference models were then used to compare the sensitivity of these models to previously reported errors in US-based collateral ligament strain estimations with their sensitivity to clinical accuracy in terms of implant positioning. The goal thereof was to assess whether the current accuracy of US-based collateral ligament strains suffices to inform computer-based pre-operative planning of TKA.

Our results showed that the knee simulator models can accurately predict tibiofemoral kinematics and loads with RMSE ranging from 0.4° to 3.8° for knee joint rotations and ranging from 0.5 mm to 2.62 mm for the translations (Table 4). For both specimens, large differences were primarily observed in anterior–posterior translation, as previously reported in the literature [29]. This is likely due to a lack of surrounding stabilizing structures in the model, such as the joint capsule, antero-lateral ligament or popliteus tendon. Nevertheless, the predicted TF kinematics displayed an overall excellent correlation and small RMSE, which compares well to prior studies using similar experimental set-ups and validation designs [29,36]. It should be noted that larger simulation errors were found with the model of specimen 2. This is probably due to the important post-TKA laxity that was observed during the ex-vivo experiments in this specimen, as displayed in Table 1, likely associated with a poorly balanced and lax LCL during the whole range of motion. Indeed, in the squatting simulation, the LCL length was found to be continuously shorter than the initial length (*L*_0_), leading to a ligament force equal to zero during the whole cycle. Because other posterolateral stabilizing structures such as the popliteofibular ligament and popliteus tendon, were not modeled, no soft-tissue structures were present to functionally contribute to the model’s lateral knee stability [37]. As a result, the model of specimen 2 displayed excessive medial laxity compared to the experimental results. Besides TF kinematics, kinetics were also compared. For both specimens, ankle and quadriceps loads were close to experimental results with RMSE percentages of the experimentally obtained forces ranging from 20% to 50% for the ankle load and ranging from 8% to 28% for the quadriceps load. For similar reasons, slightly larger differences were again observed for specimen 2. Therefore, it can be concluded that both models reproducing realistic TF kinematics and loads and can be considered valid to perform the intended sensitivity analyses.

This sensitivity analysis of the models to residual errors in US-based collateral ligament strain estimation displayed an overall limited impact on TF kinematics. Indeed, for all TF kinematics except internal–external tibial rotation, the applied alterations in MCL and LCL stiffness led to absolute variations below 0.08° and 0.1 mm, in terms of rotations and translations respectively (Table 6). The most important variations were indeed observed for the internal–external tibial rotation with values ranging from −0.61° to 0.73°. Considering the role of collateral ligaments as internal rotational stabilizers post-TKA [8], such an impact on internal–external tibial rotation was to be expected. Nevertheless, an even higher impact was expected on varus–valgus and medial–lateral translations because the collateral ligaments are the primary frontal-plane knee stabilizers post-TKA [8,9]. Likely, the impact of the stiffness perturbations is to an extent mitigated by the post and cam successfully providing medio-lateral stability in the posterior-stabilized implant design used in this study [38]. This interplay between the implant design and the soft-tissue structures in providing stability is assumed to also explain the overall smaller impact of MCL stiffness variations in specimen 2 (Table 5). Here, the excessive laxity of the second specimen likely reduced the relative importance of the collateral ligaments within this interplay. Nevertheless, the combination of these two specimens uniquely illustrates the robustness of our model with US-based collateral ligament properties to clinically occurring variations in terms of ligament balancing.

Interestingly, our models displayed a larger sensitivity of predicted TF kinematics to the current clinical inaccuracies in terms of implant positioning, more precisely to variations in internal–external rotation of the tibial baseplate of ±3.2° [34]. Once again, and as expected, the largest impact was observed on the internal–external tibial rotation with values ranging from −1.86° to 1.95°, i.e., a factor 3 times larger than the above sensitivity to errors in US-based collateral ligament strain estimation (Table 6). Interestingly, the medial–lateral translation was also found to be up to ten times more sensitive to variations in implant positioning, with values ranging from −0.70 mm to 0.70 mm (Table 6). For specimen 1, the sensitivity of varus–valgus rotation, anterior–posterior and inferior–superior translations were all very similar to the sensitivity to errors in US-based collateral ligament strain estimation. For specimen 2, increased sensitivity was also observed for varus–valgus and anterior–posterior translation. This observation further supports our hypothesis that, due to the high laxity and poor balancing of specimen 2, the implant design has a more important contribution to knee stability. By consequence, perturbations in terms of implant positioning have a greater impact on TF kinematics. Secondly, it should be noted that the above sensitivities vary throughout the flexion range. For specimen 1, the sensitivity to errors in US-based strain estimation of the LCL exceeded the sensitivity to variations in implant positioning at low flexion angles (<40° of knee flexion), as displayed in Figure 3B. This can be explained by the decreasing importance of the LCL as a medio-lateral stabilizer with increasing flexion observed in our model, because the LCL strain was found to reduce in the model. Finally, the model of specimen 1 also indicated a relatively larger sensitivity to errors in US-based strain estimation of the LCL compared to the MCL. This is a direct result of the fact that the error in US-based LCL strain estimation was also higher than in the MCLs [19]).

The main limitation of this study is the limited number of specimens. Nevertheless, this is similar to other modeling studies of the knee making use of in-vitro knee-joint simulators for validation purposes [26,29,36,39]. The second limitation is the pre-experimental conditions of both specimens used in the study. Indeed, as mentioned previously, the second specimen was poorly balanced, leading to an excessively lax specimen (Table 1). Most importantly, the fact that the LCL was inactive during squat motion rendered perturbations in LCL stiffness to reflect the errors in US-based collateral ligament strain estimation a priori useless. Nevertheless, this model was still performing well and results were deemed satisfactory to perform further sensitivity analyses to variations in MCL stiffness and implant positioning. Furthermore, as already indicated above, the combination of these two specimens allowed us to analyze the robustness of US-informed models to clinically occurring variations in terms of ligament balancing. Indeed, our model nicely reflected how well-balanced post-TKA knees rely more on collateral ligaments to guarantee functional stability during squatting, whereas poorly balanced TKA’s rely more on the implant design to ensure this stability. A third limitation is related to the definition of soft tissue structures in our computational model. Only a limited number of soft-tissue structures were included, of which insertions points were based on an anatomical atlas and not defined on a subject-specific basis. Additionally, ligaments were modeled with a limited number of straight lines only, not taking into account any wrapping of soft-tissues around the bones. Nevertheless, this generally represents a worst-case scenario in terms of the model’s sensitivity to errors in US-based collateral ligament strain estimation, and further improvements in terms of soft-tissue modelling are expected to further reduce these sensitivities. Another limitation is the optimization process, which cannot guarantee a global optimum of parameters α and β (Table 1), as well as stiffness parameters (Table 2). Nevertheless, as it was not the main goal of this study to develop optimal models but rather to study the effect within a window of variations on these parameters, this not expected to influence any of the study findings. A last limitation of this study is that only variations in internal–external rotation of the tibial baseplate of one single, cruciate sacrificing implant design were assessed. However, it should also be noted here that we opted for a worst-case scenario because we expect that adding variations in other degrees of freedom of the implant position, as well as the addition of one or both cruciate ligaments in the post-TKA model with even larger reported variations in implant malpositioning [40], would rather further increase the relative sensitivity of predicted knee kinematics to implant malpositioning in favor of our main hypothesis.

## 5. Conclusions

This study is the first of its kind to analyze the sensitivity of knee joint computer models to residual errors in ultrasound-based collateral ligament strain estimations and compare it with sensitivity to clinical accuracy in terms of implant positioning. We conclude that the overall sensitivity of the models to residual errors in ultrasound-based strain estimation is lower than the sensitivity to the current limits of accuracy in terms of implant position. This clearly further supports the potential of US-based collateral ligament strains to inform computer-based pre-operative planning of TKA. Nevertheless, one must be cautious with their use at lower knee flexion angles and future work is warranted to improve the accuracy of LCL strain estimation further.

## Figures and Tables

**Figure 1 sensors-23-08268-f001:**
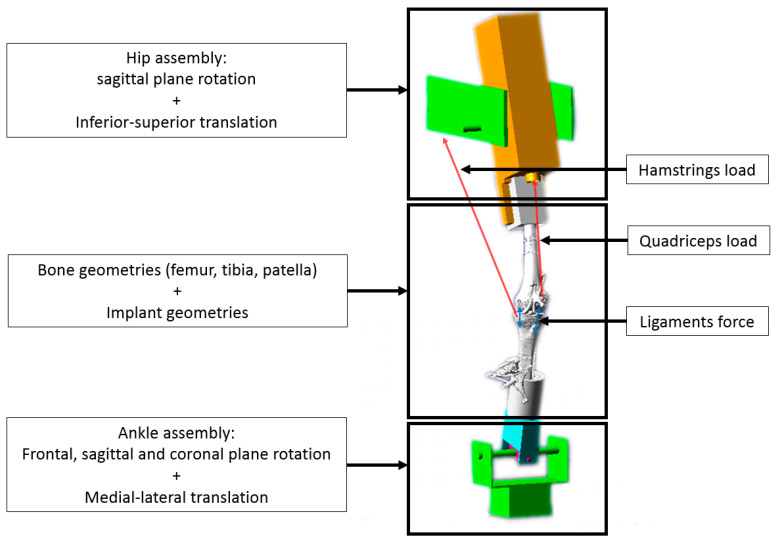
Virtual knee-joint simulator. The femur is fixed to the hip assembly free to rotate in the sagittal plane and to translate along the coronal axis). The tibia is linked to the ankle assembly with a cylindric joint (allowing rotation in the coronal plane). The ankle assembly is free to rotate in all planes and translate along the sagittal axis. Tibiofemoral and patellofemoral joints are created using a contact function constraining the joints based on the bone and implant geometries. Additionally, the model is also constrained by active loading, i.e., quadriceps load between patella and hip assembly (3 bundles) and hamstrings load between tibia and hip assembly (1 bundle medial and 1 bundle lateral) as well as passive forces from soft tissues, i.e., patellar ligament between the tibia and the patella (3 bundles) and both collateral ligaments: MCL (2 bundles) and LCL (1 bundle).

**Figure 2 sensors-23-08268-f002:**
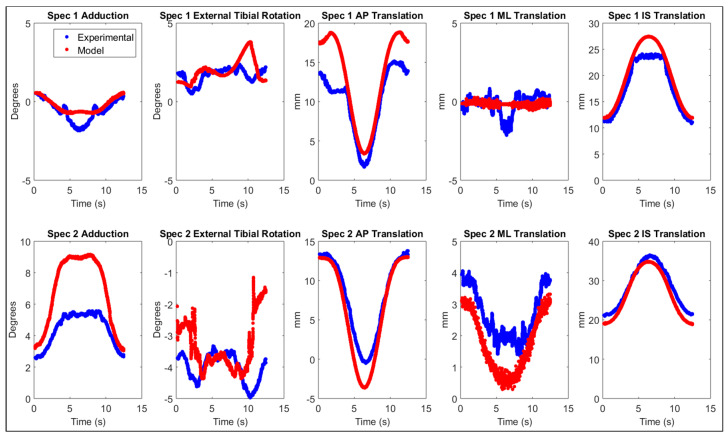
Tibiofemoral kinematics of both specimens (Specimen 1 on **top** and Specimen 2 on **bottom**) experimentally obtained (blue) and simulated with the developed model (red). The displayed kinematics are, from left to right, adduction, external tibial rotation, anterior–posterior (AP) translation, medio-lateral (ML) translation and inferior–superior (IS) translation.

**Figure 3 sensors-23-08268-f003:**
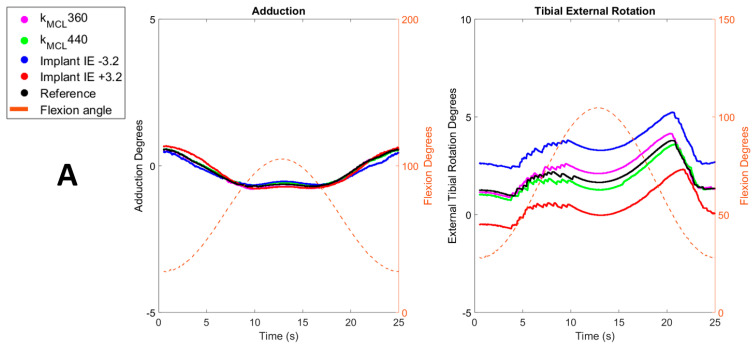
Comparisons of simulated adduction and external tibial rotation obtained after modification of ligaments stiffness (magenta and green) and results obtained after modification of implant internal rotation (blue and red) across the flexion range (dashed orange) with optimal kinematics (black). Parts (**A**,**B**) display comparisons of Specimen 1 for MCL and LCL, respectively. Part (**C**) displays comparisons of Specimen 2.

**Table 1 sensors-23-08268-t001:** Laxity of both specimens for each relevant kinematic parameter during passive flexion–extension, defined as the absolute difference between minimal and maximal value across the complete range of motion. The kinematic parameters are varus–valgus (VV), internal–external tibial rotation (IE), anterior–posterior translation (AP), medial–lateral translation (ML) and inferior–superior translation (IS).

Kinematic Range-of-Motion	Specimen 1	Specimen 2
VV (°)	5.15	10.77
IE (°)	8.37	21.00
AP (mm)	30.24	50.94
ML (mm)	3.60	8.27
IS (mm)	21.41	26.77

**Table 2 sensors-23-08268-t002:** Fine-tuned parameters used to model the passive quadriceps force.

	α	β
Specimen 1	85	0.05
Specimen 2	32.5	0.05

**Table 3 sensors-23-08268-t003:** Final ligament parameters following optimization of stiffness including reference strain ($\varepsilon_{r})$ and stiffness (k).

	Parameter	MCL	LCL	MPFL	LPFL	Patellar Ligament
Specimen 1	$\varepsilon_{r}$	0.04	0.08	0.08	0.06	−0.25
	k (N)	400	650	6000	3000	25000
Specimen 2	$\varepsilon_{r}$	0.04	0.08	0.08	0.06	−0.25
	k (N)	700	650	6000	3000	20,000

**Table 4 sensors-23-08268-t004:** Comparison of measured and computed kinematics and loads using RMSE across the complete squat cycle and Pearson correlation coefficient ρ. Load differences are also expressed as percentage of the mean experimentally obtained forces. Flexion–extension (FE), varus–valgus (VV), internal–external tibial rotation (IE), anterior–posterior translation (AP), medial–lateral translation (ML), inferior–superior translation (IS), ankle load and quadriceps load are shown.

Kinematics	Measure	Specimen 1	Specimen 2
FE (°)	RMSE	0.5	3.8
	ρ	1	0.99
VV (°)	RMSE	0.44	2.55
	ρ	0.85	0.98
IE (°)	RMSE	0.56	0.84
	ρ	0.15	−0.20
AP (mm)	RMSE	2.62	1.54
	ρ	0.96	0.99
ML (mm)	RMSE	0.50	0.92
	ρ	0	0.93
IS (mm)	RMSE	2.01	1.66
	ρ	0.99	0.99
Ankle load (N)	RMSE	17.8 (20.23%)	44 (49.96%)
	ρ	0.8	−0.11
Quadriceps load (N)	RMSE	83 (8.00%)	215 (27.77%)
	ρ	0.99	0.89

**Table 5 sensors-23-08268-t005:** Stiffness “k” of MCL and LCL to obtain the same difference in ligament strains equivalent to the error in US-based strain estimation for each specimen, i.e., ±0.27% and ±0.57% for MCL and LCL respectively [19].

	Specimen 1	Specimen 2
k_MCL_	360–440 N	650–745 N
k_LCL_	280–1250 N	/

**Table 6 sensors-23-08268-t006:** Mean differences ± standard deviation across the complete squat cycle between optimal tibiofemoral kinematics and US tibiofemoral kinematics (top) as well as mean differences across the complete squat cycle between optimal tibiofemoral kinematics and IMP tibiofemoral kinematics (bottom). Differences are defined as optimal model—US model. Varus–valgus (VV), internal–external tibial rotation (IE), anterior–posterior translation (AP), medial–lateral translations (ML) and inferior–superior translations (IS) are shown.

Kinematics	Specimen 1				Specimen 2	
Impact of MCL and LCL stiffness perturbations						
	k_MCL_ 360 N	k_MCL_ 440 N	k_LCL_ 280 N	k_LCL_ 1250 N	k_MCL_ 650 N	k_MCL_ 745 N
ΔVV (°)	−0.03 ± 0.03	0.04 ± 0.03	−0.01 ± 0.03	0.01 ± 0.04	0.08 ± 0.07	−0.06 ± 0.05
ΔIE (°)	0.26 ± 0.22	−0.29 ± 0.12	−0.61 ± 1.06	0.73 ± 0.87	−0.28 ± 0.25	0.27 ± 0.18
ΔAP (mm)	−0.02 ± 0.03	0 ± 0.30	−0.1 ± 0.19	0.1 ± 0.20	−0.05 ± 0.02	0.03 ± 0.02
ΔML (mm)	−0.02 ± 0.07	0.02 ± 0.06	0.08 ± 0.14	−0.07 ± 0.14	−0.03 ± 0.07	0.02 ± 0.07
ΔIS (mm)	0.02 ± 0.01	0.02 ± 0.01	0.05 ± 0.03	−0.02 ± 0.01	0.01 ± 0.02	0.02 ± 0.02
Impact of implant alignment perturbations						
	Implant IE − 3.2°		Implant IE + 3.2°		Implant IE − 3.2°	Implant IE + 3.2°
ΔVV (°)	0.01 ± 0.10		−0.02 ± 0.01		0.49 ± 0.12	−0.45 ± 0.12
ΔIE (°)	−1.51 ± 0.25		1.56 ± 0.22		−1.86 ± 0.42	1.95 ± 0.37
ΔAP (mm)	−0.02 ± 0.27		0.04 ± 0.17		−0.27 ± 0.14	0.22 ± 0.27
ΔML (mm)	0.70 ± 0.13		−0.70 ± 0.12		0.62 ± 0.12	−0.61 ± 0.12
ΔIS (mm)	−0.06 ± 0.23		0.06 ± 0.20		0.04 ± 0.14	0 ± 0.20

## Data Availability

The data presented in this study are available on request from the corresponding author. The data are not publicly available due to ethical and privacy considerations associated with human cadaveric donor material.

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
