# Peer review of "Sensitivity of Model-Based Predictions of Post-TKA Kinematic Behavior to Residual Errors in Ultrasound-Based Knee Collateral Ligament Strain Assessment"

_sensors, 2023, doi:10.3390/s23198268_

Round 1
Reviewer 1 Report
This paper investigated the sensitivity of post-arthroplasty kinematic predictions to ultrasound-based strain estimation errors compared to clinical inaccuracies in implant positioning. Ultrasound-derived strain residual errors were then introduced by perturbing lateral collateral ligament (LCL) and medial collateral ligament (MCL) stiffness. Afterwards, the implant position was perturbed to match with the current clinically inaccuracies reported in the literature. Finally, the impact on simulated post-arthroplasty tibiofemoral kinematics was compared for both perturbation scenarios. The article still has the following problems:
1、In Table 1 and Table 4, the title says that anterior-posterior translation is abbreviated as PD, but in the table it has been changed to AP, so it is suggested to revise it.
2. In Section 3.2 of the paper, "The average (standard deviation) difference between optimal TF kinematics and American TF kinematics is shown in Table 6".And "See Table 6 for the differences between the optimal TF kinematics and IMP TF kinematics" repeated and suggested revisions by the authors.
3. In the discussion section, please indicate the table of data extraction to help readers understand.
4. The resolution of some pictures is too low, and the text annotation in the picture needs to be enlarged.
5. What does the black line in Figure 3 represent? Please add relevant content to the article.
6. The values of article line 342 range from -0.70mm to 0.70mm. Is this data quoted in Table 6? Please double check with the author.
7. Line 357 of the article refers to your previous paper, there is a problem with the quotation, please correct it.
Reviewer 2 Report
The manuscript concerns a study on ultrasound-based knee ligament strain assessment for model-based predictions of post-total knee arthroplasty kinematics. The sensitivity of predictions to residual errors in ultrasound measurements is addressed.
In my opinion, the subject is interesting, and the manuscript is well written. However, I recommend some improvements before publication.
In particular, the manuscript lacks complete information about theoretical aspects related to the application of ultrasound measurement to the assessment of strain in knee ligaments. I suggest the Authors provide more details about the operating principle and the experimental setup, and give some explanation about the assessment of strain in ligaments from ultrasound measurements, also discussing the types and the sources of possible errors.
Some information about the optimization of the model (see Section 4,) may be useful to better understand the procedure and, possibly, to replicate it. For example, in Section 2.2 the Authors should provide information at least about the objective function and the optimization algorithm adopted.
In addition, the following minor comments could help the Authors to improve the manuscript.
· Line 139: the acronym “STL” should be explained.
· Eq. (1): explain why the linear strain limit it is set at 0.03.
· Lines 178-179: motivate why, differently from prior studies, the virtual knee-joint simulator was driven by hip displacement.
· Eq. (5): correct the position of the dot (it should be written above \delta).
· Lines 190-193: please, give some justification about the values taken from [27].
· Table 5: please, clarify the meaning of the pair of values shown in each cell (does the first value refer to an error of 0.27% or 0.57%, see line 221?)
· Line 139: the acronym “RMSE” should be explained.
Round 2
Reviewer 1 Report
The authors really improved the manuscript.
Author Response
Thank you for your suggestions which helped us improve the final manuscript.
Reviewer 2 Report
Although the Authors haven't followed all the suggested revisions, the paper in the present form, can be published provided a check on formula (5), and in particular on the apex \tau to the first \delta term
Author Response
Thank you for your suggestions. They allowed to significantly improve the quality of the final manuscript.
Regarding the equation 5, the equation is correct and written as in the literature (reference 27, p5). We would be pleased to have more information about the potential error in that equation and correct it accordingly.